# Vision-based detection and quantification of maternal sleeping position in the third trimester of pregnancy in the home setting– Building the dataset and model

Allan J. Kember[1,2,3]*, Rahavi Selvarajan[4], Emma Park[3], Henry Huang[5], Hafsa Zia[6], Farhan Rahman[4], Sina Akbarian[7], Babak Taati[5,7,8], Sebastian R. Hobson[1,9], Elham Dolatabadi[2,7]

**1** Department of Obstetrics and Gynaecology, University of Toronto, Toronto, Canada, **2** Institute of Health Policy, Management, and Evaluation, University of Toronto, Toronto, Canada, **3** Shiprah Biomedical Inc., Toronto, Canada, **4** Department of Electrical and Computer Engineering, University of Toronto, Toronto, Canada, **5** Institute of Biomedical Engineering, University of Toronto, Toronto, Canada, **6** Temerty Faculty of Medicine, University of Toronto, Toronto, Canada, **7** Vector Institute, Toronto, Canada, **8** Department of Computer Science, University of Toronto, Toronto, Canada, **9** Department of Obstetrics and Gynaecology, Maternal-Fetal Medicine Division, Mount Sinai Hospital, Toronto, Canada

* allan.kember@mail.utoronto.ca

**Data Availability Statement:** The authors, nor the owner of the data (Shiprah Biomedical Inc.), do

## Abstract

In 2021, the National Guideline Alliance for the Royal College of Obstetricians and Gynaecologists reviewed the body of evidence, including two meta-analyses, implicating supine sleeping position as a risk factor for growth restriction and stillbirth. While they concluded that pregnant people should be advised to avoid going to sleep on their back after 28 weeks' gestation, their main critique of the evidence was that, to date, all studies were retrospective and sleeping position was not objectively measured. As such, the Alliance noted that it would not be possible to prospectively study the associations between sleeping position and adverse pregnancy outcomes. Our aim was to demonstrate the feasibility of building a vision-based model for automated and accurate detection and quantification of sleeping position throughout the third trimester–a model with the eventual goal to be developed further and used by researchers as a tool to enable them to either confirm or disprove the aforementioned associations. We completed a Canada-wide, cross-sectional study in 24 participants in the third trimester. Infrared videos of eleven simulated sleeping positions unique to pregnancy and a sitting position both with and without bed sheets covering the body were prospectively collected. We extracted 152,618 images from 48 videos, semi-randomly down-sampled and annotated 5,970 of them, and fed them into a deep learning algorithm, which trained and validated six models via six-fold cross-validation. The performance of the models was evaluated using an unseen testing set. The models detected the twelve positions, with and without bed sheets covering the body, achieving an average precision of 0.72 and 0.83, respectively, and an average recall ("sensitivity") of 0.67 and 0.76, respectively. For the supine class with and without bed sheets covering the body, the models achieved an average precision of 0.61 and 0.75, respectively, and an average recall of 0.74 and 0.81, respectively.

not have the REB's nor the participants' permission to share the study data given that the study data are videos of study participants that cannot be de-identified. Should other researchers have questions about the data, they should contact the Research Ethics Manager, Health Sciences at the University of Toronto Research Ethics Board, whose contact information is available online (https://research.utoronto.ca/ethics-human-research/research-ethics-boards).

**Funding:** This study was funded by a Mitacs Accelerate Program grant (No. IT 26263). Mitacs Accelerate (Ottawa, Canada), which connects companies with over 50 research-based universities through graduate students and postdoctoral fellows, who apply their specialized expertise to business challenges. Interns transfer their skills from theory to real-world application, while the companies gain a competitive advantage by accessing high-quality research expertise. In this study, the intern was HH. The university was the University of Toronto. The professor was ED. The company was Shiphrah Biomedical Inc. (Toronto, Canada). The total study funding was $15,000 CAD and was provided through Mitacs to the University of Toronto for ED to administer for the intern (HH) and the study expenses. For the grant (No. IT 26263), Mitacs provided 50% of the study funding and had no other role in the study. Mitacs contribution to the grant (No. IT 26263) was matched by SBI, which provided the remaining 50%. SBI also provided an internship experience for HH. HH was co-supervised by ED (University of Toronto) and AJK (SBI). Mitacs had no role in study design, data collection and analysis, decision to publish, or preparation of the manuscript, whereas SBI had a role in all these aspects via AJK. Mitacs Accelerate website: https://www.mitacs.ca/en/programs/accelerate Shiphrah Biomedical Inc. website: https://shiphrahbiomedical.com.

**Competing interests:** I have read the journal's policy and the authors of this manuscript have the following competing interests: AJK is the volunteer (unpaid) Chief Executive Officer and President of one of the study funders, Shiphrah Biomedical Inc. (SBI). AJK did not receive financial or material payment for his involvement in this study. EP is a volunteer at SBI and a shareholder in SBI. EP received a financial payment for her involvement in this study from the study funds. HH was involved in this work via an internship as part of his graduate studies at the University of Toronto. HH received a financial payment for his internship from the study funds. HH has no role in ownership, management, or control of SBI. RS, HZ, FR, SA,

## Author summary

Over the last decade, mounting evidence has pointed to sleeping on the back in late pregnancy as a possible risk factor for fetal growth restriction and stillbirth. One such analysis indicated that avoiding sleeping on the back in late pregnancy may be as important as stopping smoking and more important than achieving a healthy pre-pregnancy weight. However, a major problem with previous research has been that a pregnant person's sleeping position was not actually measured but was, rather, recollected by the person themselves, which is known to be inaccurate. Currently available sleep position measurement devices, including computer vision based algorithms, are not appropriate for use in pregnancy. As such, there is a need for tools to help researchers measure sleeping position in pregnancy. Using an ordinary home-surveillance camera, we collected night-vision video of pregnant participants simulating sleeping positions unique to pregnancy in the home setting and in the presence and absence of bed sheets. We built a model from this data to automatically and accurately detect sleeping position and measure the amount of time spent in each position. Our model may eventually contribute to a tool that could help bridge a major gap in previous research, which would strengthen the case for a definitive and universal change in clinical practice.

## Introduction

Sleep during pregnancy plays a crucial role in maternal and fetal complications as evidenced by prior studies [1–11]. A supine going-to-sleep position after 28 weeks' gestation was shown to be associated with more than double the odds of late stillbirth [12–17], and more than triple odds of birthing a small-for-gestational age (SGA) infant [12–15,18]. The going-to-sleep position is the dominant sleep position for the entire night in more than 50% of pregnancies with most pregnant people spending an average of 44% of the night in their sleep onset position [19]. Analyses demonstrate that avoiding the supine going-to-sleep position is important for all pregnancies after 28 weeks' gestation regardless of risk factors for stillbirth in the pregnant person (e.g., age, body size, smoking, recreational drug use, pre-existing hypertension or diabetes) or in the fetus (e.g., movements, growth, gestation) [17].

In high-income countries, late stillbirth, defined as stillbirth occurring at or greater than 28 weeks' gestation, remains an important yet understudied public health issue, affecting 1.3–8.8 per 1000 births [20]. Sleeping position in pregnancy, if a truly causative risk factor for stillbirth, must be contextualized among other modifiable risk factors. The leading three potentially modifiable risk factors for stillbirth are maternal obesity, advanced maternal age, and smoking–all of which have been shown to be deleterious to placental health and fetal growth [21]. Of these, only smoking is a realistically modifiable factor during the course of pregnancy, which amplifies the potential importance of sleeping position as a potentially modifiable risk factor [22–24]. For comparison, consider that if all pregnant people avoided sleeping on the back in the third trimester of pregnancy, the overall late stillbirth rate could be reduced by 5.8% (population attributable risk), [17] whereas if all pregnant people quit smoking, the overall late stillbirth rate would be reduced by 5.5% [25]. Furthermore, for a given pregnant person who usually goes to sleep on their back, by going to sleep on their left side instead, they can reduce their risk of late stillbirth by 62% (aOR 0.38) [17], whereas one Canadian study demonstrated that a 10% reduction in pre-pregnancy body weight would reduce the risk of stillbirth by only 10% [26].

BT, SRH, and ED have no competing interests that could be perceived to bias this work.

The potential contribution of supine sleeping position in late pregnancy to stillbirth and growth restriction is biologically plausible. When in the supine position after 28 weeks' gestation, the inferior vena cava (IVC) is compressed by the gravid uterus and pregnancy, resulting in reduction in blood flow and oxygen delivery to the fetus [27–33]. Furthermore, the prevalence of supine sleeping position in the third trimester is common with a wide range of variability–published research of objective sleeping positions demonstrate that pregnant people spend between 9.5–47% of the night sleeping supine [19,22,23,34–39].

The National Institute for Health and Care Excellence (NICE) organization's recently reviewed the evidence for safe sleeping positions in pregnancy and concluded that, on a balance, the evidence is strong enough to recommend that pregnant people should be advised to avoid going to sleep on their back after 28 weeks' due to the likely link with stillbirth [40]. Indeed, the aforementioned studies demonstrating an association between supine sleep, stillbirth, and delivering an SGA infant were all retrospective and body position was subjectively measured through self-reporting. As such, these studies are limited by recall bias, established inaccuracies inherent in self-reported sleep behavior when compared to objective measurements [35,41–45], and by the fact that sleeping position is usually not static in healthy sleep nor pregnancy [22,46,47]. Furthermore, members of our group have previously demonstrated that pregnant people underestimate the percentage of time they spend sleeping supine by an average of 7% in absolute terms (44% underestimate in relative terms) [22]. Among the NICE's main critiques of the evidence was that it would not be possible to study the association between sleeping position in pregnancy and adverse outcomes because of these limitations [40]. As such, herein lays the rationale for the present study.

Over the past decade, advances in computer vision have revolutionized daily life, and deep neural networks in particular, have attained remarkable performance in object detection tasks [48,49]. There have been numerous studies utilizing computer vision technologies (CVT) to track and analyze body movement and posture [50–55]; however, none of these were developed for pregnancy, which has unique sleep behaviors and physiology. Non-CVT devices for body position detection such as BodyCompass [56] and SleepPos [57–60] were also not developed with pregnancy in mind. These devices focus on the position of the thorax. While the position of the thorax may be important for maternal obstructive sleep apnea (OSA) diagnosis and treatment in pregnancy and the postpartum [19,61–64], the position of the pelvis, owing to its impact on maternal and uteroplacental hemodynamics [27,29,65–68], is more important to fetal wellbeing.

The literature concerning investigations of the impact of OSA on fetal heart rate (FHR) patterns is of questionable quality in that it rarely accounts for maternal posture, a major confounder, using the gold standard (video determination). While OSA and accompanying maternal oxygen desaturations may play a role in spontaneous nocturnal FHR decelerations [69–71], this likely only holds true for moderate or severe OSA with deep desaturations and concomitant placental disease (e.g., fetal growth restriction) [70,71]. The fetus has a remarkable adaptive capacity to withstand in utero hypoxia, and the fetal hemoglobin has a higher affinity for oxygen than the maternal adult hemoglobin. Evidence from the largest studies to date investigating the role of OSA indicates no relationship between mild maternal oxygen desaturations and FHR decelerations, even in pregnancies with known fetal growth restriction [71,72]. Conversely, objectively-verified supine maternal posture seems to play a role in overnight FHR patterns, and this is corroborated by several studies [19,23,29], and likely extends to adverse pregnancy outcomes [17,18]. Only one study showed no effect of supine maternal sleeping position on FHR patterns; however, this study did not confirm maternal position with video, and the median percentage of time spent in the supine position overnight was only 1.09%, which makes it challenging to draw definitive conclusions [73].

Further, determination of only the thorax position is of insufficient resolution for sleep-in-pregnancy research because discordance between the position of the pelvis and the position of the thorax naturally occurs when the body is in a hybrid (i.e., twisted) position–in one of our studies (ClinicalTrials.gov Identifier: NCT04437407), we have observed up to 144 minutes of discordance between the pelvis and thorax position and up to 26.5 minutes of discordance between the pelvis and head position in a single night, which exposes a major limitation of use of "thoracic-centric" and/or "head-centric" position detection devices in sleep-in-pregnancy research.

In the present study, we built a model for effortless, accurate, unobtrusive, and non-contact detection and measurement of sleeping position, at a high resolution, occurring throughout the third trimester of pregnancy in the home setting.

## Materials & methods

### Design

This study can be conceptualized as a cross-sectional design in which video data of simulated sleeping positions was prospectively collected over a period of five to ten minutes. No methodological changes were made after the study's commencement.

### Patient and public involvement

Development of the research question and outcome measures, design of the study, recruitment process, and conduct of the study were completed by the researchers without the involvement of the public or patients. Participants in this study indicated whether they wished to receive a summary of the overall study results during the informed consent process.

### Participants

Participants were recruited Canada-wide by the researchers via various social media platforms (Instagram, Facebook, LinkedIn, and Twitter) and an app for research study participation (Honeybee). Snowball sampling was also used in which participants who completed their participation in the study invited their friends and family members who also may have been eligible to participate in contacting the researchers. All contact between the researchers and participants was virtual (i.e., no in-person or face-to-face contact given the ongoing COVID-19 pandemic). All data collection was completed within the participants' own homes.

Participants were eligible to participate in the study if they were healthy (self-reported), had a low-risk singleton pregnancy, were in the third trimester of pregnancy (between 28 weeks' and 0 days through 40 weeks and 6 days gestation, inclusive, by first-trimester ultrasound), were aged 18 to 50 years, had access to an Android or iPhone device, and had a secure 2.4 GHz network (WiFi or cellular data plan) in their home. Exclusion criteria included non-English speaking, reading, or writing and a musculoskeletal condition preventing simulation of sleeping positions.

### Interventions

The consent process was completed via a virtual meeting (telephone or Zoom), and eligible participants gave voluntary, written, informed consent to participate in the study. Basic demographic data was collected from each participant including, name, mailing address, age, gestational age, height, and weight. No additional personal health information was accessed or shared because linkage to clinical outcomes was beyond the scope of this study. Each participant was then mailed a study kit containing sanitized study equipment.

A virtual meeting was held with each participant to describe the specific study protocol and instructions including assisting in equipment setup and data collection. Each participant installed an infrared home-surveillance camera (Wyze Cam V2 by Wyze Labs, Inc., Seattle, USA) by attaching it to the wall, centered at the head of their bed and 1.6 to 1.7 meters above the sleeping surface, to clearly capture all pertinent data. The participant then shared the video feed of the Wyze Cam with the researcher via the Wyze smartphone app, and the researcher provided instructions in real-time to the participant. The participant was assisted in taking two short video clips of themselves in twelve different simulated body positions. We used a simulated position approach to data collection to create a highly heterogeneous dataset for CVT model building. See the supplementary file S1 Appendix for a justification of why recordings of natural sleep were not used.

Each participant simulated the body positions while lying in their bed in the dark and fully clothed. The participant was asked to use a bed sheet with no patterns and remove all pillows and objects from the bed except for head pillows. The participant simulated each of the twelve positions for ten seconds per position (recorded at 10 frames per second, fps), first without bed sheets covering her body, and subsequently with bed sheets covering her body. While simulating each position, the participant was asked to move her head, arms, and legs constantly while maintaining the position with her body. The individual who appears in the images in this manuscript has given written informed consent (as outlined in PLOS consent form) to publish their photos. The twelve simulated positions, eleven of which are sleeping positions and one of which is a transitional position, were (see Fig 1): left recovery (P1), left lateral (P2), left tilt (P3), supine (P4), supine thorax with left pelvic tilt (P5), supine thorax with right pelvic tilt (P6), supine pelvis with left thorax tilt (P7), supine pelvis with right thorax tilt (P8), right tilt (P9), right lateral (P10), right recovery (P11), and sitting up at the edge of the bed (P12). All

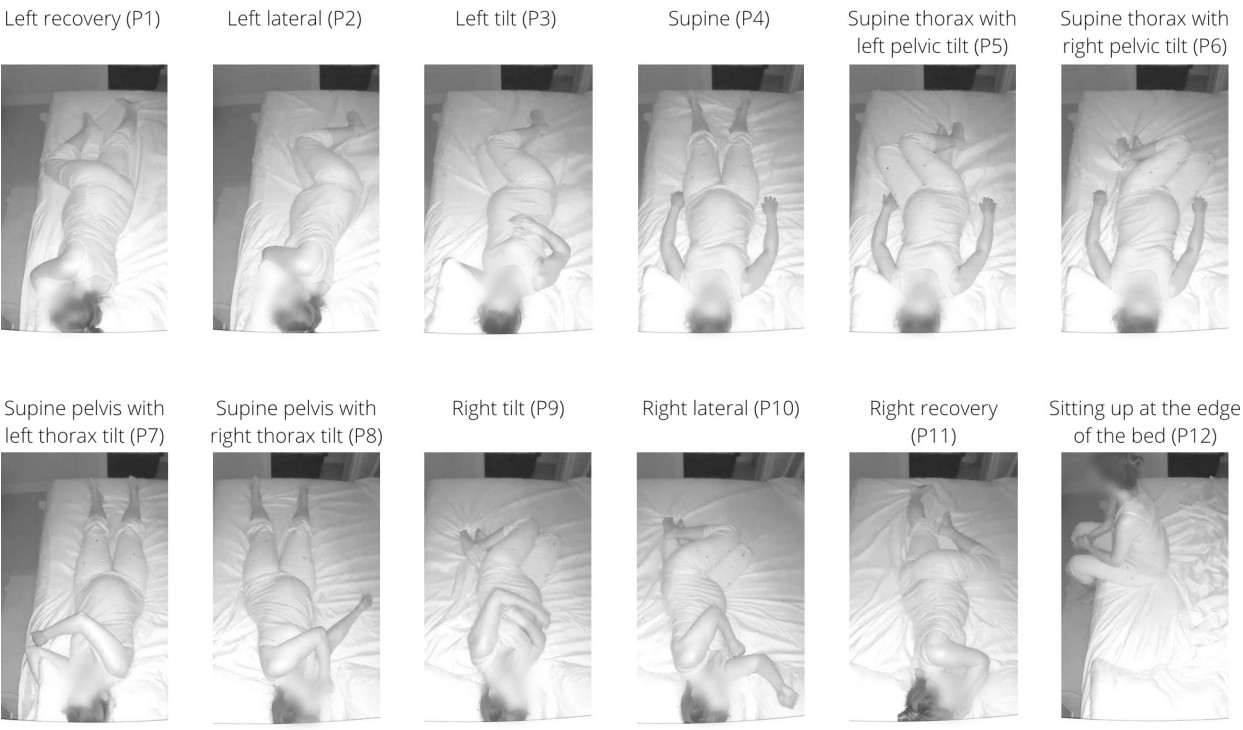

**Fig 1. Twelve simulated positions.** Example of each of the twelve simulated positions, P1 through P12, without a bed covering the participant's body.

"tilt" positions were tilted in the lateral direction (i.e., rotated about a string connecting the head to the feet) and were simulated at approximately 45° above the horizontal plane–to achieve this angle, participants were instructed to tilt their body "45°, that is, half-way between completely on your side and flat on your back". When the simulation was complete, each participant uploaded her video clips to the research team's secure cloud server for analysis and storage.

### Outcomes

Not applicable.

### Sample size

A formal sample size calculation was not performed. We determined a target sample size of N = 20, which was informed by the study budget and timeline.

### Statistical methods

We assessed normality of continuous variables via the Shapiro-Wilk test at a 0.05 significance level. Normally distributed continuous variables are presented as "mean ± standard deviation", and non-normally distributed continuous variables are presented as "median (interquartile range)". Statistical analyses were conducted using the R Statistical Software package (R version 4.1.1 (2021-08-10) — "Kick Things") [74].

### Computer vision model development

**Dataset development and annotation.**  First, all video recordings were converted to a sequence of frames at a rate of 10 fps using an open-source video and audio processing software, FFmpeg [75], (refer to Fig 2). All extracted frames were then manually reviewed and sorted into the twelve unique postures, shown in Fig 1, with and without bed sheets (24 classes in total).

Next, all low-quality frames were excluded from the dataset. Low-quality frames included frames in which there was static posing (no movement between subsequent frames), minimal movement between subsequent frames, unidentifiable postures, transitions between postures, low image resolution, and glitches in the video recording.

Moreover, frames in which there was no participant in the field of view (i.e., the object being detected was not in the frame), which we refer to as "negative frames", were set aside to be used as negative frames (background) in the training dataset [76].

The remaining frames were semi-randomly down-sampled such that there were approximately 10 frames per class per participant for annotation. Annotation included localization of the participant's body within the frame and labeling of the body posture occurring in the frame. An open-source annotation tool, LabelImg [77], was used to localize the participant's body in each frame via a rectangular bounding box that touched the edges of their body. As such, the development dataset to be used for model training and evaluation included the annotated (labeled) dataset and the negative frames (non-labeled).

**Model development and evaluation.**  We framed the detection of sleeping position in bed as a multi-class classification (24 classes) where state-of-the-art deep learning Convolutional Neural Networks were used. In this regard, we used YOLOv5s with 7.5M parameters [78,79] including pre-trained weights from the COCO dataset and fine-tuned it on our annotated dataset to make predictions of classes. We trained our deep learning models on an NVIDIA

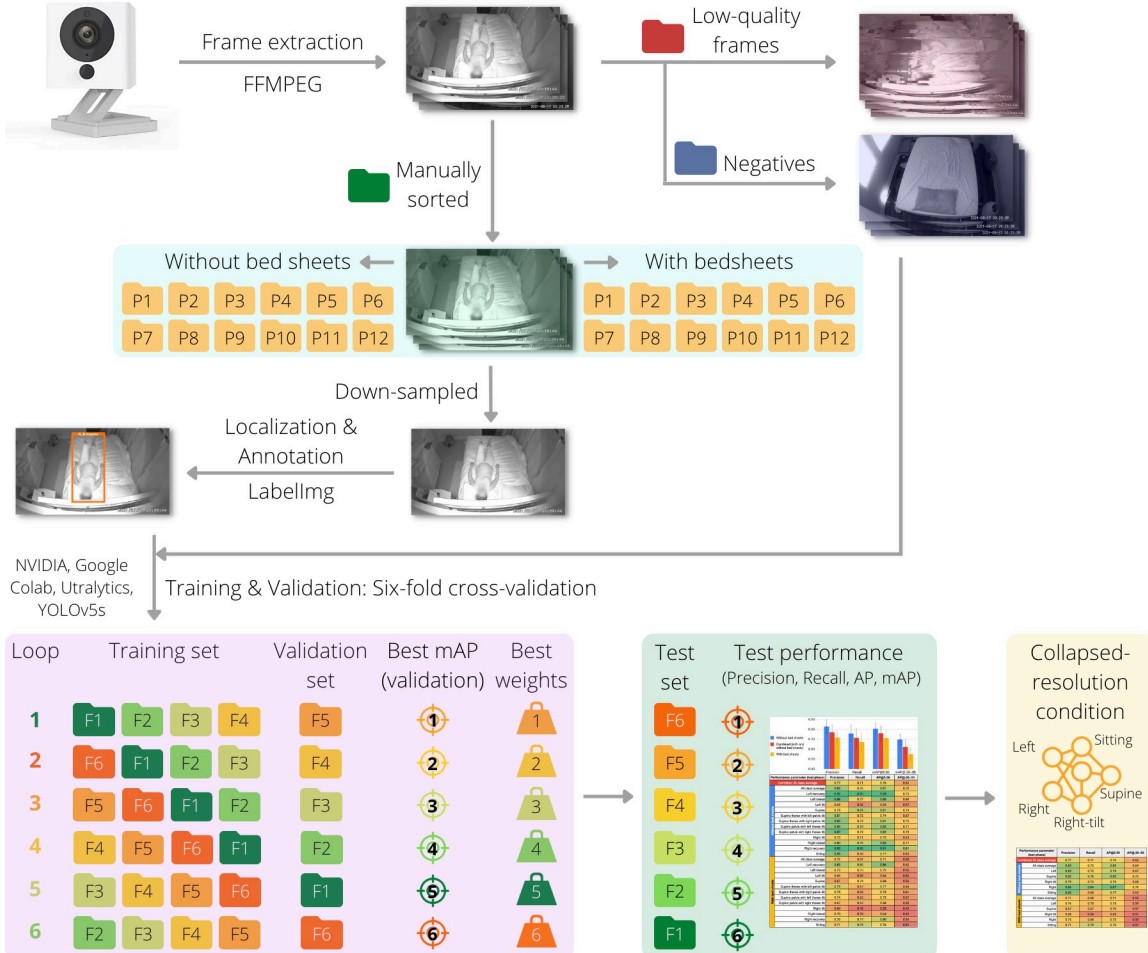

**Fig 2. Schematic representation of dataset development and annotation and model development and evaluation.** Frames were extracted from video recordings via FFMPEG and manually reviewed to classify them into the twelve positions without bed sheets and with bed sheets covering the body. Low-quality frames and frames without a participant (negatives) were set aside. The manually sorted frames were then semi-randomly down-sampled and localization and annotations were then completed with LabelImg. The negative frames were augmented using the ImageDataGenerator class in the Keras library and were then combined with the annotated frames in the training and validation phase, which was by six-fold cross-validation on a virtual GPU (NVIDIA Tesla P100 on Google Colab) using YOLOv5s by Ultralytics and pre-trained weights from the COCO dataset. The weights of the model in each loop of the cross-validation that achieved the best mAP on the validation set were saved as the best weights. Performance evaluation for each loop was completed on the testing set using the best weights, and the performance was evaluated by computing the mAP, precision, and recall. Various classes with similar anatomic and hemodynamic implications on uteroplacental perfusion were subsequently combined, which resulted in a simpler, collapsed-resolution (CR) condition (see the supplementary file S2 Appendix).

Tesla P100 GPU virtual machine running on Google Colab Pro Plus using the PyTorch framework with a batch size of 16 and early stopping criteria [80,81].

During training, we disabled "flip augmentation" in the YOLO configuration file because we wanted our model to distinguish left and right sleeping positions as separate classes. The clinical rationale for distinguishing between left and right as separate classes is that while the human body looks symmetrical on the outside, it is not anatomically symmetrical on the inside, which has physiologic implications. On the inside, the relatively thick-walled and high-pressure aorta runs down the left side of the spinal column whereas the relatively thin-walled and low-pressure IVC runs up the right side of the spinal column. As such, the left lateral and tilt positions are not likely hemodynamically equivalent to the right lateral and tilt positions,

and tilting from the supine position to the right tilt position may actually worsen the compression of the IVC (rather than relieving it like tilting to the left does) and reduces cardiac output, and this effect is supported by several studies [82–87].

Using k-fold cross-validation, the participants and pertaining frames were randomly split into six non-overlapping folds (k = 6) of equal size (equal number of participants). In each loop of the cross-validation, four folds were used for training ("training set"), one fold was used for validation ("validation set"), and one fold was held out for testing ("testing set"). For each training loop, the weights that achieved the highest mean average precision (mAP) estimate across all classes on the validation set were saved. Six loops of the cross-validation procedure were completed such that each fold was given a turn to be the validation set and testing set. Finally, performance evaluation was completed using the best weights on the testing set for each loop, and precision, recall, and average precision (AP) at different intersections over union (IoU, described below) were calculated and compared across each class.

The precision for a class is the precision average across all recall (sensitivities) values between 0 and 1 at various IoU thresholds. The IoU is the area of overlap between the predicted object location and the ground truth location divided by the area of union (the total area of both the prediction and ground truth combined). During object detection, the IoU is calculated to determine whether the object is correctly detected or not. By interpolating across all points, the AP can be interpreted as the area under the curve of the precision-recall curve. For each class, we also report the AP@0.50 and the AP@.50-.95. The 0.50 in "AP@0.50" is the IoU threshold value. If the predicted IoU is greater than or equal to the threshold, it is considered a true positive. As such, the AP@0.5 means that the AP is calculated for the objects predicted with an IoU value of 0.50 or greater. The AP@.50-.95, similar to the previous definition, means the AP is calculated by considering multiple threshold values between 0.50 and 0.95 and averaging them across all those thresholds. For a given class, a very high value for AP@.50-.95 indicates that the model is more confident about its predictions. The mean average precision (mAP) across all classes was calculated by taking the mean of the AP's across all classes being predicted by the model.

See the supplementary file S1 Appendix for more details regarding the rationale for a simulated posture study design, bias-variance trade-off in k-fold cross-validation, selection of YOLOv5s, early stopping criteria, and definitions of precision and recall.

In addition, we presented the performance evaluation of the models from each loop under a collapsed resolution (CR) condition where we reduced the total number of classes by combining classes with similar anatomic and hemodynamic implications from the perspective of uteroplacental perfusion (see the supplementary files S2 Appendix and S1 Fig).

## Ethics

This study was approved by the University of Toronto Health Sciences Research Ethics Board in June 2021 (Protocol No. 27465). Written informed consent was given by all study participants.

## Results

From July 5th, 2021 through September 22nd, 2021, 39 participants were assessed for eligibility. Of these, 12 (31%) were not recruited (2 gave birth prior to giving written informed consent, 1 was excluded due to twin pregnancy, and the remaining 9 declined participation)–we did not collect any data from these. Of the 27 (69%) participants that met the eligibility criteria and gave written informed consent to participate in the study, 24 (89%) successfully completed

**Table 1. Demographic characteristics of participants (n = 24) included in the computer vision model.**

| Characteristic | Value |
|---|---|
| Age (years) | 29.0 (26.8–31.3) |
| **At time of data collection (video)** | |
| Gestational age (weeks') | 33.7 ± 3.4 |
| Height (meters) | 1.64 (1.63–1.65) |
| Weight (kilograms) | 77.7 ± 14.7 |
| BMI (kg/m$^2$) | 27.9 ± 5.1 |

the study. Two (7%) gave birth prior to completing data collection, and one (3%) was excluded by the researchers due to irremediable technical difficulties.

## Demographic characteristics

Demographic characteristics of the 24 participants included in our model are shown in Table 1. Age and height were not normally distributed and had medians of 29 years and 1.64 meters, respectively.

## Dataset

We collected 257.12 minutes of infrared 1080P video at 10 frames-per-second from which we extracted a total of 152,618 frames. Of these, 82,251 frames (53.9%, 137.33 minutes) were recorded with no bed sheets covering the body, and 70,367 (46.1%, 119.78 minutes) were recorded with bed sheets covering the body. Of the total 152,618 frames, 88,793 (58.2%) were low-quality frames and 206 (0.1%) were negative frames, which left 63,619 (41.7%) frames for annotation with the 24 classes. Of these 63,619 frames, 33,190 (52.2%) frames were with no bed sheets and 30,429 (47.8%) were with bed sheets. See Table 2 for the number of frames extracted and manually sorted for each class without bed sheets and with bed sheets.

## Models

Our development set included 5970 annotated frames and 206 negative frames. We used six-fold cross validation for model training and evaluation. See Table 3 for a summary of the

**Table 2. Total number of frames extracted for each class.**

| Class description | Without bed sheets | With bed sheets |
|---|---|---|
| | Number of frames | Number of frames |
| Left recovery | 2562 | 2799 |
| Left lateral | 3333 | 2491 |
| Left tilt | 3130 | 2602 |
| Supine | 2937 | 3302 |
| Supine pelvis with left thorax tilt | 3527 | 2569 |
| Supine pelvis with right thorax tilt | 2689 | 2980 |
| Supine thorax with left pelvic tilt | 2486 | 2318 |
| Supine thorax with right pelvic tilt | 1944 | 1794 |
| Right tilt | 2735 | 2020 |
| Right lateral | 2625 | 2948 |
| Right recovery | 2534 | 1986 |
| Sitting up at the edge of the bed | 2688 | 2620 |
| Remaining frames (low quality, negatives) | 49061 | 39938 |

**Table 3. Summary of validation and performance parameters (validation precision, validation recall, validation mAP@0.50, and validation mAP@.50-.95) during training and validation of six models via six-fold cross-validation, across 24 classes (12 classes with bed sheets covering the body, 12 class without bed sheets covering the body).**

| Loop of CV | Number of validation frames | Number of epochs completed | Best results at epoch # | Validation Precision | Validation Recall | Validation mAP@0.50 | Validation mAP@.50-.95 |
|---|---|---|---|---|---|---|---|
| 1 | 985 | 84 | 63 | 0.708 | 0.800 | 0.811 | 0.665 |
| 2 | 882 | 150 | 129 | 0.759 | 0.803 | 0.839 | 0.648 |
| 3 | 967 | 119 | 98 | 0.780 | 0.767 | 0.814 | 0.652 |
| 4 | 967 | 81 | 60 | 0.727 | 0.778 | 0.779 | 0.594 |
| 5 | 985 | 118 | 97 | 0.749 | 0.782 | 0.813 | 0.685 |
| 6 | 952 | 108 | 87 | 0.803 | 0.780 | 0.812 | 0.649 |

CV = cross-validation; mAP@0.50 = mean average precision at an intersection of union threshold of 0.50; mAP@.50-.95 = mean average precision averaged across multiple intersections of union thresholds between 0.5 and 0.95.

training and validation, which is given for reference purposes. The summary includes, for each loop of the cross-validation, the number of validation frames, the number of training epochs completed prior to activation of the early stopping criteria, the epoch with the best results (highest validation mAP@0.50), and the performance parameters (validation precision, validation recall, validation mAP@0.50, and validation mAP@.50-.95) corresponding to the best results and across the 24 classes (12 classes with bed sheets covering the body, 12 classes without bed sheets covering the body).

Table 4 provides a summary of the testing of each of the six models produced from the six-fold cross-validation. These six models are defined by the weights achieving the highest validation mAP@0.50 during the training and validation phase. For each model, the number of testing frames in the testing set, number of testing instances, and performance parameters (testing precision, testing recall, testing mAP@0.50, and testing mAP@.50-.95) across 24 classes (12 classes with bed sheets covering the body, 12 class without bed sheets covering the body) are presented.

Fig 3 shows an elaboration of the results in Table 4 by presenting a grouped bar chart, and Fig 4 shows these results via a heat map. In Fig 3, the grouped bar chart shows the four performance parameters–precision, recall, mAP@0.50, and mAP@.50-.95 –from the testing phase across the six models' testing sets and across all classes under three different conditions: (1) Red bar: a combination of the "with bed sheets" and "without bed sheets" conditions (24 classes). (2) Blue bar: the "without bed sheets" condition (12 classes). (3) Yellow bar: the "with bed sheets" condition (12 classes). The error bars represent one standard deviation of the respective value across all measures, which reflects the variability both across models (n = 6) and classes

**Table 4. Summary of testing and performance parameters (testing precision, testing recall, testing mAP@0.50, and testing mAP@.50-.95) during testing of the six models produced by six-fold cross-validation, across 24 classes (12 classes with bed sheets covering the body, 12 class without bed sheets covering the body).**

| Model | Number of testing frames | Number of instances | Testing Precision | Testing Recall | Testing mAP@0.50 | Testing mAP@.50-.95 |
|---|---|---|---|---|---|---|
| 1 | 938 | 882 | 0.806 ± 0.128 | 0.757 ± 0.159 | 0.798 ± 0.100 | 0.633 ± 0.105 |
| 2 | 1008 | 952 | 0.822 ± 0.119 | 0.779 ± 0.182 | 0.823 ± 0.105 | 0.693 ± 0.104 |
| 3 | 1041 | 985 | 0.778 ± 0.162 | 0.736 ± 0.161 | 0.766 ± 0.144 | 0.663 ± 0.147 |
| 4 | 978 | 922 | 0.759 ± 0.112 | 0.743 ± 0.181 | 0.776 ± 0.124 | 0.629 ± 0.135 |
| 5 | 982 | 926 | 0.735 ± 0.210 | 0.570 ± 0.193 | 0.669 ± 0.177 | 0.537 ± 0.162 |
| 6 | 1023 | 967 | 0.726 ± 0.159 | 0.700 ± 0.180 | 0.721 ± 0.162 | 0.578 ± 0.160 |

Performance parameters given as mean ± standard deviation. mAP@0.50 = mean average precision at an intersection of union threshold of 0.50; mAP@.50-.95 = mean average precision averaged across multiple intersections of union thresholds between 0.5 and 0.95.

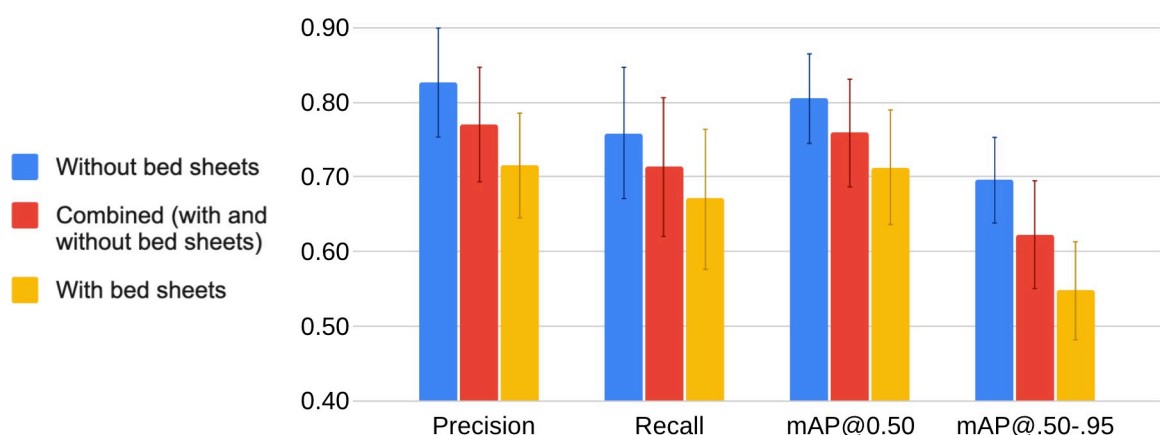

**Fig 3. Grouped bar charts of performance parameters from the testing phase (precision, recall, mAP@0.50, and mAP@.50-.95) averaged across the six models' testing sets and across all classes, where the parameters are presented under three conditions.** (1) Red bar: averaged across all classes as a combination of the "with bed sheets" and "without bed sheets" conditions (24 classes). (2) Blue bar: averaged across all classes under the "without bed sheets" condition (12 classes). (3) Yellow bar: averaged across all classes under the "with bed sheets" condition (12 classes). The error bars represent one standard deviation of the respective value across all measures, which reflects the variability both across models and classes.

(n = 24 for combined condition; n = 12 for the without and with bed sheets conditions). In Fig 4, the heatmap shows the four performance parameters (columns)–precision, recall, AP@0.50, and AP@.50-.95 –from the testing phase averaged across the six models' test sets for each of the predicted classes (rows) under the "without bed sheets" and "with bed sheets" condition. The "all class average" is provided as the averaged value of the respective performance parameter across the six models' test sets and the 12 classes under each bed sheets condition, and the "combined 24 class average" is given as the average of the former two values combined.

On a per-condition basis, higher and more consistent values of performance were generally achieved under the "without bed sheets" condition, whereas the values for "with bed sheets" conditions were lower and had higher variability. On a per-class basis under the "without bed sheets" condition, the recovery classes had higher values ($\geq$0.90) of AP@0.50, and the lateral, supine, and supine hybrid classes had values between 0.80 to 0.90, whereas non-hybrid tilted (left tilt, right tilt), sitting, and supine thorax with left pelvic tilt generally had lower (<0.80) AP@0.50 as averaged across the six models. When the "with bed sheets" condition is considered, the recovery classes maintained higher values ($\geq$0.80) of AP@0.50, whereas the left lateral, supine hybrid, and sitting classes had values between 0.70 to 0.80, and the right lateral, non-hybrid tilted (left tilt, right tilt), supine, and supine pelvis with right thorax tilt classes generally had lower (<0.70) AP@0.50 as averaged across the six models. An enduring pattern with regard to the AP@0.50 across models for the "without bed sheets" and "with bed sheets" conditions is that the recovery classes scored high and non-hybrid tilted (left tilt, right tilt) classes scored low.

Fig 5 displays a running example using one of our trained models to localize and classify the sleeping position of a study participant. As shown in the figure, the participant's body on the bed is localized by the colored bounding box. Moreover, the prediction of the sleeping position along with its confidence score (between 0 and 1) are shown at the top of each bounding box.

## Harms

There were no known harms related to this study. The time spent supine and in positions with a supine pelvis, while simulated in both video clips for each participant, was minimal. No participants reported symptoms of supine hypotensive syndrome.

| Performance parameter (test phase) | | Precision | Recall | AP@0.50 | AP@.50-.95 |
|---|---|---|---|---|---|
| Combined 24 class average | | 0.77 | 0.71 | 0.76 | 0.62 |
| Without bed sheets | All class average | 0.83 | 0.76 | 0.81 | 0.70 |
| | Left recovery | 0.92 | 0.90 | 0.93 | 0.75 |
| | Left lateral | 0.88 | 0.77 | 0.85 | 0.68 |
| | Left tilt | 0.69 | 0.56 | 0.65 | 0.57 |
| | Supine | 0.75 | 0.81 | 0.81 | 0.74 |
| | Supine thorax with left pelvic tilt | 0.81 | 0.72 | 0.74 | 0.67 |
| | Supine thorax with right pelvic tilt | 0.85 | 0.73 | 0.81 | 0.73 |
| | Supine pelvis with left thorax tilt | 0.84 | 0.78 | 0.83 | 0.71 |
| | Supine pelvis with right thorax tilt | 0.87 | 0.75 | 0.82 | 0.73 |
| | Right tilt | 0.73 | 0.71 | 0.72 | 0.63 |
| | Right lateral | 0.80 | 0.79 | 0.83 | 0.71 |
| | Right recovery | 0.93 | 0.90 | 0.91 | 0.81 |
| | Sitting | 0.85 | 0.68 | 0.77 | 0.63 |
| With bed sheets | All class average | 0.72 | 0.67 | 0.71 | 0.55 |
| | Left recovery | 0.83 | 0.80 | 0.86 | 0.62 |
| | Left lateral | 0.73 | 0.75 | 0.72 | 0.55 |
| | Left tilt | 0.69 | 0.59 | 0.64 | 0.54 |
| | Supine | 0.61 | 0.74 | 0.68 | 0.56 |
| | Supine thorax with left pelvic tilt | 0.79 | 0.67 | 0.77 | 0.64 |
| | Supine thorax with right pelvic tilt | 0.75 | 0.63 | 0.73 | 0.61 |
| | Supine pelvis with left thorax tilt | 0.74 | 0.62 | 0.73 | 0.57 |
| | Supine pelvis with right thorax tilt | 0.67 | 0.65 | 0.68 | 0.58 |
| | Right tilt | 0.60 | 0.45 | 0.53 | 0.42 |
| | Right lateral | 0.70 | 0.59 | 0.64 | 0.43 |
| | Right recovery | 0.76 | 0.77 | 0.80 | 0.56 |
| | Sitting | 0.71 | 0.79 | 0.76 | 0.51 |

**Fig 4. Heatmap of precision, recall, AP@0.50, and AP@.50-.95 (columns) from the testing phase averaged across the six models' test sets for each of the predicted classes (rows) under the "without bed sheets" (upper blue row header) and "with bed sheets" condition (lower yellow row header).** The value of the respective performance parameter is mapped to a color spectrum from red to yellow to green where values of 0.50 or less are represented by red at the lower end of the spectrum, values around 0.75 are shades around yellow (oranger if lower than 0.75; greener if higher than 0.75), and values of 0.90 or more are represented by green at the higher end of the spectrum. The "all class average" is provided as the averaged value of the respective performance parameter across the six models' test sets and the 12 classes under each bed sheets condition, and the "combined 24 class average" is given (red column) as the average of the former two values combined. For the "all class average" rows, the value in the AP@0.50 column is the mAP@0.50, and the value in the AP@.50-.95 column is the mAP@.50-.95 since these values represent averages across multiple classes.

## Discussion

There exists a significant need to study sleeping positions in the third trimester of pregnancy using objective methods. These methods enable quantifying the time ("dose") spent in various sleeping positions across this trimester and linking these "doses" to pregnancy outcomes ("responses"). Data on this dose-response is valuable and may lend support to or detract support from previous associations found between supine sleep, stillbirth, and small-for-gestation infants. The data underlying these previous associations are of suboptimal quality because they are based on subjective self-reported recollection of sleeping position and do not account for intra- and inter-night variability in sleeping position [17,18,40]. Currently, there are few

## Ground Truth

## Prediction

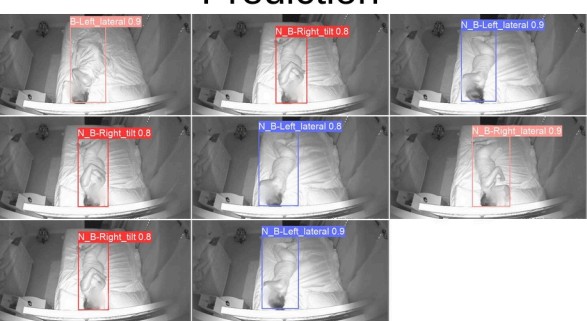

**Fig 5. Example output of one of our trained models to localize and classify the sleeping position of a study participant.** Ground truth localizations and labels are on the left. Model predictions of localizations and labels are on the right.

clinically-approved products to monitor people's sleep positions; however, given the unique physiology and sleep behaviors characterizing pregnancy, these products are not appropriate for use in sleep-in-pregnancy research.

To advance the field of sleep-in-pregnancy efficiently, there is a need for simple, readily accessible research tools that can be easily and widely employed in the home setting by the participant without the need for a research assistant to visit the participant's home. Herein, we have demonstrated that models can be developed to automatically process any infrared video (e.g., from any available home-surveillance camera) as long as the camera is positioned above the head of the bed. These tools should be accurate. The American Academy of Sleep Medicine's Manual for the Scoring of Sleep and Associated Events states that body position reporting is "required" for polysomnographic studies and "optional" for home-based studies; however, there is currently no consensus on how accurate body position reporting should be. Despite training our models on less than 10% of our collected dataset, our models achieved an overall mAP@0.50 of 0.76 (0.81 without bed sheets, 0.71 with bed sheets; see Fig 4) in the testing phase on unseen images across 24 classes. The tools must account for factors unique to pregnancy anatomy and physiology including, but not limited to, the impact of the pelvis position on uteroplacental hemodynamics and fetal physiology, the location of the low-pressure, thin-walled, and collapsible IVC on the right side of the spine, and the challenge that pregnancy places on the respiratory system, especially during sleep [19,38], and especially when supine [19,38,88,89]. The models we developed may meet this requirement–we were able to train them to detect sleeping position at a high resolution in that it accounts for the position of the pelvis and the thorax and the direction (right vs. left) at varying degrees (recovery, lateral, tilt) for a total of eleven sleeping postures. The tools, if CVT-based, must be able to compensate for the natural sleeping environment including the presence or absence of a bed partner, bed sheets, pillows, and dark or low-light conditions, which our model partially accounts for but is discussed later as a major limitation of our study. With few exceptions, all performance parameters for detected classes in the absence of bed sheets were equal to or greater than the respective value in the presence of bed sheets (see Fig 4), which was not unanticipated given the expected negative effect on performance of bed sheets obscuring the participant's body. Sleep in the third trimester is punctuated by frequent position changes due to discomfort and frequent rising to void due to decreased bladder capacity from compression by the gravid uterus–our models can account for this and is trained to detect sitting at the edge of the bed, which signifies entry/exit events.

Over the last few years, several vision-based sleeping posture classification studies have been published including Grimm et al. (2016) [90], Liu et al. (2017) [91], Mohammadi et al.

(2018) [92], Wang et al. (2019) [93], Li et al. (2022) [94], and Akbarian et al. (2019) [54]. Like Mohammadi et al. [92] and Li et al. [94], we used video from a readily available home-security camera, simulated sleeping position data, and incorporated the presence and absence of bed sheets. Mohammadi et al. [92] used a CNN algorithm (as we did), trained and validated their model using approximately 10,000 images, and completed resampling via leave-one-out (LOO) cross-validation (one participant was "held out", and a model was trained on the remaining participants, and this was repeated for each participant). Note that LOO cross-validation is a form of k-fold where k (the number of folds) is set equal to the number of samples, n. Similarly, we used a cross-validation approach to resampling but with k-fold cross-validation (with k = 6). We were able to calculate the average recall ("average sensitivity") from Mohammadi et al.'s testing data as 0.64 (SD 0.10) and 0.79 (SD 0.09) with and without bed sheets, respectively, which is comparable to the average recall across our six models at 0.67 (0.08) and 0.76 (SD 0.09) with and without bed sheets, respectively (see Fig 4). Comparison of Li et al.'s [94] testing performance to ours is not possible because they only present classification accuracy (not recall, precision, nor mAP) but a direct comparison of accuracy is not possible because we cannot compute it for our results–accuracy computation requires the number of true negatives [95], which is not computed by YOLO, which is a detector and does not generate negative predictions. Furthermore, standard research and clinical scoring of body position is limited to left, right, supine, prone, or upright [96]; however, for physiologic reasons unique to pregnancy, our models score body position at a higher resolution by accounting for body twist and hybrid positions (e.g., supine thorax with left pelvic tilt). As such, even if there was consensus on how accurate body position reporting should be, this would need to be extrapolated as eight of our models' eleven detected sleeping positions are non-standard positions.

Our study has two main limitations that prevent our model from being ready for deployment in the real-world. The first limitation is related to our models' generalization performance due to the biases in the video data. The range of our participants' BMI's was relatively narrow and clustered in the normal and overweight category, which limits generalizability to pregnancies with higher BMI categories (e.g., class II obesity), which are common and a specific concern vis-à-vis sleeping position. Our models were built on video data that included only one subject in the bed, which limits our models' generalizability for multi-subject posture detection. Simultaneous multi-subject posture detection is important given the high prevalence (78–85%) of co-sleeping in pregnancy [22,24]. Moreover, our participants were asked to use thin, pattern-free bed sheets and to clear their beds of any pillows and objects except for head pillows, which is not realistic of the natural sleeping environment in pregnancy where it is common to use multiple pillows (in addition to head pillows) to support one's body against while sleeping; therefore, it is anticipated that the performance of our model would deteriorate with patterned and/or thick bed sheets (e.g., duvet) and in the presence of a bed partner and multiple pillows. In addition, we did not account for the prone posture; however, note that the occurrence of prone posture during natural sleep in the third trimester is exceedingly rare due to increasing abdominal size [22,23,34–38]. Lastly, our study was relatively small with only 24 participants. As such, our model could benefit from increased heterogeneity in participants and their environments. Currently, we are collecting a real-world video dataset of sleep in pregnancy, which will address each of these issues.

The second limitation of this study is related to data protection and privacy considerations with long-term recording of continuous nocturnal video data in the home setting. This issue limits the translation of our models into research (and possibly clinical) practice. A viable and more likely alternative to continuous long-term use is periodic sampling (e.g., recording one night per week throughout the third trimester), which should yield a good surrogate of sleeping position across the third trimester.

## Supporting information

**S1 Appendix. Additional methodological details.**
(PDF)

**S2 Appendix. Collapsed-resolution model.**
(PDF)

**S1 Fig. Heatmap of precision, recall, AP@0.50, and AP@.50-.95 (columns) from the testing phase averaged across the six models for each of the predicted collapsed-resolution (CR) classes (rows) under the "without bed sheets" (upper blue row header) and "with bed sheets" condition (lower yellow row header).** The value of the respective performance parameter is mapped to a colour spectrum from red to yellow to green where values of 0.50 or less are represented by red at the lower end of the spectrum, values around 0.75 are shades around yellow (oranger if lower than 0.75; greener if higher than 0.75), and values of 0.90 or more are represented by green at the higher end of the spectrum. The "all class average" is provided as the averaged value of the respective performance parameter across the six models' test sets and the five collapsed-resolution classes under each bed sheets condition, and the combined "24 class average" is given (red column) as the average of the former two values combined. For these "all class average" rows, the value in the AP@0.50 column is a mAP@0.50, and the value in the AP@.50-.95 column is a mAP@.50-.95 since these values represent averages across multiple classes.
(TIF)

## Acknowledgments

The authors would like to acknowledge and thank Ms. Alexandra Gratton for her assistance in design and promotion of recruitment advertisements for this study on social media.

## Author Contributions

**Conceptualization:** Allan J. Kember, Henry Huang, Sina Akbarian, Babak Taati, Sebastian R. Hobson, Elham Dolatabadi.

**Data curation:** Allan J. Kember, Emma Park, Henry Huang, Hafsa Zia.

**Formal analysis:** Allan J. Kember, Rahavi Selvarajan, Emma Park, Henry Huang, Hafsa Zia, Sina Akbarian, Elham Dolatabadi.

**Funding acquisition:** Allan J. Kember, Henry Huang, Elham Dolatabadi.

**Investigation:** Allan J. Kember, Henry Huang, Elham Dolatabadi.

**Methodology:** Allan J. Kember, Rahavi Selvarajan, Henry Huang, Farhan Rahman, Sina Akbarian, Babak Taati, Sebastian R. Hobson, Elham Dolatabadi.

**Project administration:** Allan J. Kember, Elham Dolatabadi.

**Resources:** Allan J. Kember, Elham Dolatabadi.

**Software:** Allan J. Kember, Rahavi Selvarajan, Farhan Rahman, Elham Dolatabadi.

**Supervision:** Allan J. Kember, Sina Akbarian, Babak Taati, Sebastian R. Hobson, Elham Dolatabadi.

**Validation:** Allan J. Kember, Rahavi Selvarajan, Elham Dolatabadi.

**Visualization:** Allan J. Kember, Elham Dolatabadi.

**Writing – original draft:** Allan J. Kember.

**Writing – review & editing:** Allan J. Kember, Rahavi Selvarajan, Emma Park, Henry Huang, Hafsa Zia, Farhan Rahman, Sina Akbarian, Babak Taati, Sebastian R. Hobson, Elham Dolatabadi.

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
