## [Decision Letter · Decision Letter 0]

20 Apr 2023

PDIG-D-22-00350

Vision-based detection and quantification of maternal sleeping position in the third trimester of pregnancy in the home setting – building the dataset and model

PLOS Digital Health

Dear Dr. Kember,

Thank you for submitting your manuscript to PLOS Digital Health. After careful consideration, we feel that it has merit but does not fully meet PLOS Digital Health's publication criteria as it currently stands. Therefore, we invite you to submit a revised version of the manuscript that addresses the points raised during the review process.

Please submit your revised manuscript within 60 days Jun 19 2023 11:59PM. If you will need more time than this to complete your revisions, please reply to this message or contact the journal office at digitalhealth@plos.org. Please include the following items when submitting your revised manuscript:

We look forward to receiving your revised manuscript.

Kind regards,

Heather Mattie

Academic Editor

PLOS Digital Health

Journal Requirements:

1. We ask that a manuscript source file is provided at Revision. Please upload your manuscript file as a .doc, .docx, .rtf or .tex.

Additional Editor Comments (if provided):

Please address all comments and suggestions noted by Reviewers.

Reviewers' comments:

Reviewer's Responses to Questions

**Comments to the Author**

1. Does this manuscript meet PLOS Digital Health’s publication criteria? Is the manuscript technically sound, and do the data support the conclusions? The manuscript must describe methodologically and ethically rigorous research with conclusions that are appropriately drawn based on the data presented.

Reviewer #1: Partly

Reviewer #2: Yes

2. Has the statistical analysis been performed appropriately and rigorously?

Reviewer #1: I don't know

Reviewer #2: Yes

3. Have the authors made all data underlying the findings in their manuscript fully available (please refer to the Data Availability Statement at the start of the manuscript PDF file)?

Reviewer #1: No

Reviewer #2: No

4. Is the manuscript presented in an intelligible fashion and written in standard English?

Reviewer #1: Yes

Reviewer #2: Yes

5. Review Comments to the Author

Reviewer #1: This study reports on an attempt to assess body position during sleep using video imaging and computer technology. The goal is to fill the gap in this technology by including pregnant people in order to better assess sleep position and pregnancy prospectively. The need to fill the scab is adequately established and the modeling is detailed and extensive. My primary concern with the study is the possibility of generalization to real world views. The fact that they required no additional pillows in the sleeping area set up an unrealistic sleep environment for the pregnant person. In the third trimester, the majority of pregnant people support their bodies in various positions using a combination of traditional pillows, body pillows, and sleep wedges. These additions may make this technology no longer functional, and I would have liked to see at least some attempts at incorporating these sleep aids in the videos. 

1. Author summary, line 46: use the phrase “sleeping on the back” but the more common term used in the literature is “supine sleep”. You could consider replacing this for consistency.

2. Introduction, line 76–77: there are other risk factors with far larger effect sizes and then many of these, such as pregestational diabetes, lupus, and previous pregnancy morbidity. Consider limiting this to a list of potentially modifiable factors. I appreciate, however, that you acknowledge that weight and body size is not realistically modifiable for the majority of people.

3. Introduction, lines 81–88: this paragraph could be moved to the discussion.

4. Introduction, lines 1 27–1 43: the majority of this could also be moved to discussion. 

5. Materials and methods, line 194–195: this is incorrect. You did, in fact, collect personal health information as names and addresses are part of personal health information.

6. Materials and methods, lines 211–212: this is my main challenge and accepting the importance of this paper. Removal of all pillows and objects from the bed except for head pillows is not consistent with how the majority of their trimester pregnant people sleep. Please either include an arm in which people do have pillows and wedges or justify the exclusion of this. 

7. Materials and methods, line 306: this is a typo; the IVC runs up the right side of the spinal column. 

8. Results, table one: this is a fairly narrow and low range of BMIs. A limitation of your study is also generalizability to hire VMI, which is common in pregnancy and is a specific concern with regards to sleep position. 

9. Discussion, lines 494–495: this is conjecture and not clear from what you present. 

10. Discussion, lines 538–545: I would emphasize these limitations more strongly and included acknowledgment that many pregnant people sleep in a bed full of support pillows. Also acknowledge limited BMI range.

Reviewer #2: This article presents an automated analysis technique to identify sleep position in pregnancy. This is a worthwhile tool to develop, due to the lack of data on changes in sleep position in late pregnancy, where supine sleep is a modifiable risk factor for stillbirth. Participants were videoed sleeping in their own homes, but on beds with light unpatterned sheets (or no sheets) and with no pillows interfering with the field of view. 

The motivation methods and results are well written and clear. However, given the results are limited by the set up of the participant imaging I would expect more detailed comment on the possible loss ability of the method to detect sleeping positions with these simplifying factors removed. Patterned sheets or a co-sleeper would reduce accuracy, are there possible (more advanced?) techniques that could be tried in the future to remedy this?

Related to this is the need for a clearer discussion on what kind of accuracy might be acceptable for the algorithms to meet their purpose in clinical and research studies. It is clear that the accuracy in your method is comparable to other methods for detecting sleep position, but what would be an acceptable accuracy for this type of study for this type of algorithm to replace, say, manual identification of sleep position? How close is your algorithm to this desired accuracy?

The article refers to a website that will in the future allow researchers to conduct similar analyses. When clicking on the weblink the website states “Coming august 2023” – if the website is not currently available, it can’t be reviewed and I question whether the link should be provided.

6. PLOS authors have the option to publish the peer review history of their article (what does this mean?). If published, this will include your full peer review and any attached files.

**Do you want your identity to be public for this peer review?** For information about this choice, including consent withdrawal, please see our Privacy Policy.

Reviewer #1: No

Reviewer #2: No

---

## [Decision Letter · Decision Letter 1]

17 Aug 2023

Vision-based detection and quantification of maternal sleeping position in the third trimester of pregnancy in the home setting – building the dataset and model

PDIG-D-22-00350R1

Dear Dr. Kember,

We are pleased to inform you that your manuscript 'Vision-based detection and quantification of maternal sleeping position in the third trimester of pregnancy in the home setting – building the dataset and model' has been provisionally accepted for publication in PLOS Digital Health.

Best regards,

Chrystinne Oliveira Fernandes

Academic Editor

PLOS Digital Health

Reviewer Comments (if any, and for reference):

Reviewer's Responses to Questions

**Comments to the Author**

1. If the authors have adequately addressed your comments raised in a previous round of review and you feel that this manuscript is now acceptable for publication, you may indicate that here to bypass the “Comments to the Author” section, enter your conflict of interest statement in the “Confidential to Editor” section, and submit your "Accept" recommendation.

Reviewer #1: (No Response)

Reviewer #2: All comments have been addressed

2. Does this manuscript meet PLOS Digital Health’s publication criteria? Is the manuscript technically sound, and do the data support the conclusions? The manuscript must describe methodologically and ethically rigorous research with conclusions that are appropriately drawn based on the data presented.

Reviewer #1: Yes

Reviewer #2: Yes

3. Has the statistical analysis been performed appropriately and rigorously?

Reviewer #1: Yes

Reviewer #2: Yes

4. Have the authors made all data underlying the findings in their manuscript fully available (please refer to the Data Availability Statement at the start of the manuscript PDF file)?

Reviewer #1: Yes

Reviewer #2: No

5. Is the manuscript presented in an intelligible fashion and written in standard English?

Reviewer #1: Yes

Reviewer #2: Yes

6. Review Comments to the Author

Reviewer #1: 1.Title: I appreciate the changes in language to emphasize this is a simulation and not evaluation of actual sleep. I suggest changing the title to reflect this, for example “Vision-based detection and quantification of simulated maternal sleeping position in the third trimester of pregnancy – building the dataset and model.”

2.Introduction, lines 78-80: I think I was unclear with my previous comment about this section. I am realizing that by “leading” you may mean “most common in the population.” Chronic HTN and diabetes have larger effect sizes and could be argued to be leading in that sense.

3.Introduction, lines 83-90: as you explain in the paragraph lines 100-109, the data on sleep position and stillbirth is mostly retrospective and limited. Because of this, I would put qualifiers on your calculations of attributable risk. The causality of this relationship is weak.

4.Introduction – it is not clear that the authors have notably edited the introduction to be more brief/concise. It remains quite long and includes details that could be in the discussion, as previously mentioned.

Reviewer #2: The authors have addressed my concerns. Thank you for making these revisions.

7. PLOS authors have the option to publish the peer review history of their article (what does this mean?). If published, this will include your full peer review and any attached files.

**Do you want your identity to be public for this peer review?** For information about this choice, including consent withdrawal, please see our Privacy Policy.

Reviewer #1: No

Reviewer #2: No
